# TGRL: Teacher Guided Reinforcement Learning Algorithm for POMDPs

**Idan Shenfeld, Zhang-Wei Hong, & Pulkit Agrawal**
Massachusetts Institute of Technology
USA
{idanshen,zwhong,pulkitag}@mit.edu

**Aviv Tamar**
Technion Institute of Technology,
Israel
avivt@technion.ac.il

## Abstract

In many real-world problems, an agent must operate in an uncertain and partially observable environment. Due to partial information, a policy directly trained to operate from these restricted observations tends to perform poorly. In some scenarios, during training more information about the environment is available, which can be utilized to find a superior policy. Because this privileged information is unavailable at deployment, such a policy cannot be deployed. The *teacher-student* paradigm overcomes this challenge by using actions of privileged (or *teacher*) policy as the target for training the deployable (or *student*) policy operating from the restricted observation space using supervised learning. However, due to information asymmetry, it is not always feasible for the student to perfectly mimic the teacher. We provide a principled solution to this problem, wherein the student policy dynamically balances between following the teacher's guidance and utilizing reinforcement learning to solve the partially observed task directly. The proposed algorithm is evaluated on diverse domains and fares favorably against strong baselines.

## 1 Introduction

In many sequential decision-making problems, the agent has uncertain or incomplete information about the system's state. This scenario is common in many real-world problems and applications, such as robotics, natural language processing, and healthcare (Kurniawati, 2022; Das et al., 2017). Developing effective policies in such environments remains a significant area of research.

Our objective is to learn policies that can function under the restricted information available during deployment. However, during training, we may have access to more information such as additional observations when training in simulation Lee et al. (2020), or accurate measurements by instrumentation during training Levine et al. (2015). This additional information can be regarded as privileged information, available only during training. The introduction of this privileged information can simplify the problem, as it enables the agent to possess all relevant information necessary for making informed decisions. For example, navigating a building using a map is more straightforward than only using a first-person view. Teacher-student learning exploits this idea by training a teacher policy, capable of solving the problem with privileged information. The teacher policy is used to guide the training of a student policy that can only access restricted observations. Teacher guidance typically improves the learning process, yielding stronger student policies than training without guidance.

In recent years, advancements in technology have made simulations an increasingly popular environment for training RL agents. The ability to control the environment and access privileged information in simulations makes them well-suited for teacher-student learning. Utilization of simulations in combination with this paradigm has been shown to be successful in a wide range of problems such as autonomous driving (Codevilla et al., 2018; Chen et al., 2020; Hawke et al., 2020), robotic locomotion (Lee et al., 2020; Margolis et al., 2022) and dexterous manipulation (Chen et al., 2022b). This is the setting considered in this work.

When looking at teacher-student learning, a straightforward approach is to train the student to mimic the teacher's actions, a method known as Imitation Learning. The idea is that if the recent history of student observations is sufficient to infer the privileged information that elicited the teacher's action,

then by conditioning the student policy on this history, it can accurately imitate the teacher even without access to the privileged information.(Kumor et al., 2021; Swamy et al., 2022)

Although the appealing simplicity of this approach, it is not always possible for the student to perfectly imitate the teacher. As an example, let's look at the "Tiger Door" environment (Figure 1) from (Littman et al., 1995; Warrington et al., 2021). A robot must navigate to the goal cell (green), without touching the failure cell (blue). The cells, however, randomly switch locations every episode, and their nature is not observed by the agent. The maze also includes a pink button that reveals the correct goal location. A teacher policy that has access to the correct goal location will be able to go to it directly. An optimal student, however, must *deviate* from the teacher's route to explore the pink button – behavior that cannot be learned by imitation. More broadly, there are environments where

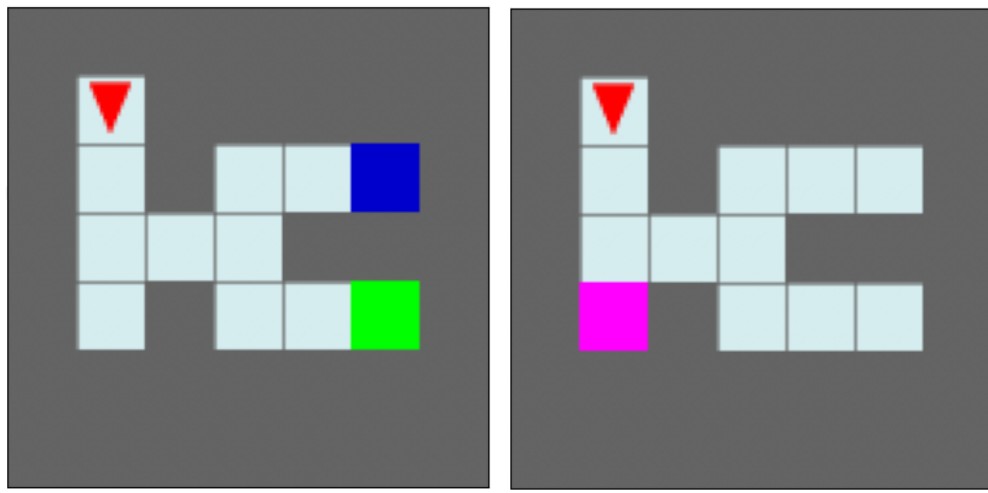

Figure 1: The Tiger Door environment. On the left is the teacher's observation, where the goal cell (in green) and the failure cell (in blue) are perceptible. On the right is the student's observation, where these cells are not visible, but there is a pink button; touching it reveals the other cells.

even when considering history, it is not enough to eliminate the difference between the student and the teacher. Trying to mimic the teacher will create a sub-optimal agent, an issue termed *Imitation Gap*. In these cases, the student should not rely solely on the teacher's advice and needs to combine learning from another signal, such as a reward, to solve the task at hand. These environments have been investigated in a series of recent papers (Weihs et al., 2021; Nguyen et al., 2022) that proposed algorithms to solve this problem. However, these algorithms rely on tuning sensitive hyperparameters to control the balance between following the teacher and learning to solve the task directly from the reward signal. Finding the right value for these parameters is not a trivial task since they are a function of the Imitation Gap in the environments, which is hard to estimate a priori. Therefore, their approaches require an exhaustive hyperparameter search to work.

In this work, we present Teacher Guided Reinforcement Learning (TGRL). A teacher-student algorithm that is suitable for problems with and without Imitation Gap. We construct a constrained optimization problem that dynamically adjusts this aforementioned balance based on the problem at hand. We test our algorithm on a series of tasks with Imitation Gap, demonstrating that our algorithm achieves comparable or superior results compared to prior work, without a need for hyperparameter tuning. Finally, we use our algorithm to solve a robotic hand re-orientation problem, using only tactile sensors to estimate the object's pose. This is considered a difficult partial observation problem due to the sensors' high-dimensional and sparse nature, demonstrating our method's usefulness.

## 2 PRELIMINARIES

**Reinforcement learning (RL).** We consider the interaction between the agent and the environment as a discrete-time Partially Observable Markov Decision Process (POMDP) (Kaelbling et al., 1998) consisting of state space $\mathcal{S}$, observation space $\Omega$, action space $\mathcal{A}$, state transition function $\mathcal{T} : \mathcal{S} \times \mathcal{A} \to \Delta(\mathcal{S})$, reward function $R : \mathcal{S} \times \mathcal{A} \to \mathbb{R}$, observation function $\mathcal{O} : \mathcal{S} \to \Delta(\Omega)$, and initial state distribution $\rho_0 : \Delta(\mathcal{S})$. The environment is initialized at an initial state $s_0 \sim \mathcal{S}$ sampled from $\rho_0$. For each timestep $t$, the agent observes the observation $o_t \sim O(.|s_t), o_t \in \Omega$, takes action $a_t$ determined by the policy $\pi$, receives reward $r_t = R(s_t, a_t)$, transitions to the next state $s_{t+1} \sim \mathcal{T}(.|s_t, a_t)$, and observes the next observation $o_{t+1} \sim O(.|s_{t+1})$. The goal of RL (Sutton & Barto, 2018) is to find the optimal policy $\pi^*$ maximizing the expected cumulative rewards (i.e., expected return). As the rewards are conditioned on states unobservable by the agent, seminal work (Kaelbling et al., 1998) has shown that the optimal policy may depend on the history of observations $\tau_t : \{o_0, a_0, o_1, a_1...o_t\}$, and not only on the current observation $o_t$. Overall we aim to find the optimal policy $\pi^* : \tau \to \Delta(A)$ that maximizes the following objective:

$$\pi^* = \arg\max_\pi J_R(\pi) := \mathbb{E}\left[\sum_{t=0}^\infty \gamma^t r_t\right]. \tag{1}$$

**Teacher-student learning.** Solving directly for the optimal policy of a POMDP is an intractable problem (Madani et al., 1999; Papadimitriou & Tsitsiklis, 1987), and even solving with deep reinforcement learning methods is shown to be a complicated task (Zhu et al., 2017). As stated before, although the agent only has access to the observation space during deployment, this is not always the same during training. When training in simulations, we can get access to another observation space, $\Omega_p$, with privileged information observations $\omega$. In teacher-student learning, we construct a teacher policy $\bar{\pi}$ that successfully solves the task over $\Omega_p$, achieving a high cumulative reward. In this work, we do not assume about the origin of $\bar{\pi}$, whether it be from RL training or some other algorithm, such as trajectory optimization. Given such a teacher policy, our goal is to use it to train a student policy to solve the task over the original observation space $\Omega$.

In Imitation Learning, we teach the student by minimizing a statistical distance function between the teacher and the student's actions. For stochastic policies, it is common (Czarnecki et al., 2019) to use the cross-entropy as the loss function resulting in the following optimization problem:

$$\max_\pi J_E(\pi) := \max_\pi \mathbb{E}\left[-\sum_{i=0}^H \gamma^t H_t^X(\pi|\bar{\pi})\right] \tag{2}$$

Where $H_t^X(\pi|\bar{\pi}) = -\mathbb{E}_{a\sim\pi(\cdot|\tau_t)}[\log \bar{\pi}(a|\omega_t)]$. When looking at this objective, a natural question arises: What is the policy we get from maximizing this objective? To answer that, we will cite the following result:

**Proposition 2.1.** *In the setting described above, denote $\pi^{IL} = \arg\max_\pi J_E(\pi)$ and $f(\omega) : \Omega_p \to \Omega$ as the function that maps observations with privileged information to observations without such information. Then, for any $\omega \in \Omega_p$ with $o = f(\omega)$, we have that $\pi^{IL}(o) = \mathbb{E}[\bar{\pi}(s)|o = f(\omega)]$.*

*Proof.* See (Weihs et al., 2021) proposition 1 or (Warrington et al., 2021) theorem 1. □

Intuitively, this implies that the student will learn the statistical average of the teacher's actions for each observable state. This approach does not account for the fact that different underlying states may require different actions. As a result, the student policy may only be able to approximate the teacher's actions by taking the average of all observed actions, which can result in a sub-optimal policy when considering the environmental reward (Eq. 1). In the Tiger Door environment, for example, the student will be able to follow the teacher's action until the second intersection. There, the teacher will go each time to the side that leads to the goal. The student, however, does not see the goal and, therefore, will have an equal probability to go to each side - an average of the teacher's actions. This policy is sub-optimal since the student will get to the goal cell only half the time. In general, these differences between the student and the teacher are termed Imitation Gap:

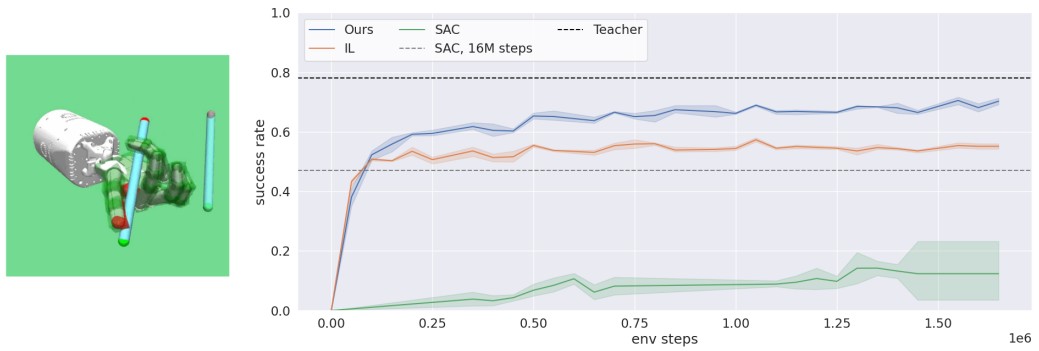

Figure 2: Success rate of a pen reorientation task by Shadow Hand robot, using tactile sensing only. While vanilla reinforcement learning takes a long time to converge, and imitation learning methods can suffer from Imitation Gap, our algorithm is able to solve the task with reasonable sample efficiency.

**Definition 2.2.** *Imitation Gap.* In the setting described above, let $\pi^*$ be the solution to Eq. 1 and $\pi^{IL}$ be be the solution to Eq. 2 (as described in proposition 2.1). Then, we term the following quantity as Imitation Gap: $J_R(\pi^*) - J_R(\pi^{IL})$.

In scenarios where the information available to the teacher is more comprehensive than that accessible to the student, policies trained through imitation learning may exhibit a non-zero Imitation Gap, as observed in the Tiger Door environment. As our objective is to find the optimal policy, $\pi^*$, imitation learning alone is insufficient in such environments.

## 3 METHODS

Although Imitation Learning cannot solve the problem in environments where Imitation Gap exists, we would still want to use the teacher's guide to help the learning process of the student. Prior work (Czarnecki et al., 2019; Nguyen et al., 2022) has suggested combining the two objectives. This way, the student will try to optimize for the environmental reward but, at the same time, will prefer actions taken by the teacher. The fact that we are also trying to maximize the environmental reward can lead to the necessary deviation from the teacher's actions since it will result in an overall higher objective value. Looking back to the Tiger Door example, following the teacher's guidance can lead you to the areas where the goal can be found, giving the agent some understanding of the environment, but a small deviation (going to the button) will result in a much higher average reward. The combined objective is:

$$\max_{\pi} J_{TG}(\pi, \alpha) = \max_{\pi} \mathbb{E}\left[\sum_{t=0}^{H} \gamma^t(r_t - \alpha H_t^X(\pi|\bar{\pi}))\right] \tag{3}$$

Where $\alpha$ is a balancing coefficient. Notice that this objective is indeed a combination of the previous two, as $J_{TG}(\pi, \alpha) = J_R(\pi) + \alpha J_E(\pi)$. This objective can also be seen as a sort of reward shaping, where the agent gets a negative reward for taking actions that differs from the teacher's action.

As the balancing coefficient between the environmental reward and the teacher guidance, the value of $\alpha$ greatly impacts the algorithm's performance. A low $\alpha$ will limit the guidance the student gets from the teacher, resulting in a long convergence time. A high value of $\alpha$ will lead to the student relying too much on the teacher, with the risk of the final policy having an Imitation Gap. A common practice is to conduct an extensive hyperparameter search to find the best $\alpha$, as different values are best suited for different tasks (Schmitt et al., 2018; Nguyen et al., 2022). Besides the inefficiency of such search, as the agent progresses on a task, the amount of guidance it needs from the teacher can vary, and a constant $\alpha$ may not be optimal throughout training. The exact dynamics of this trade-off are task-dependent, and per-task tuning is tedious, undesirable, and often computationally infeasible.

### 3.1 TEACHER GUIDED REINFORCEMENT LEARNING

To reduce the effect of $\alpha$ on the optimization problem, we want to find another mechanism that will make sure that the policy we learn will be the optimal policy of the POMDP. To achieve this, we will add a constraint to our optimization problem that will enforce this goal. Hence, our optimization problem becomes:

$$\max_{\pi} J_{TG}(\pi, \alpha) \quad \text{s.t} \quad J_R(\pi) \geq \eta \tag{4}$$

Where $\eta \in \mathbb{R}$. By adding this constraint, we limit the set of feasible policies to those whose cumulative reward (Eq. 1) is at least $\eta$. If we would choose $\eta = J_R(\pi^*)$, then this set will include only the optimal policies. Unfortunately, $J_R(\pi^*)$, the value of the optimal policy, is usually not known beforehand. To overcome this, we will introduce an auxiliary policy, $\pi_R$, and will use its value as the lower bound $\eta = J_R(\pi_R)$. This way, we ensure that the student policy will be at least as good as a policy trained without the teacher. Overall, Our algorithm iterates between improving the auxiliary policy by solving $\max_{\pi_R} J_R(\pi_R)$ and solving the constrained problem using Lagrange duality. We will note that a recent paper (Chen et al., 2022a) uses a similar constraint in another context, to adjust between exploration and exploitation in conventional RL. More formally, for $i = 1, 2, \ldots$ we iterate between two stages:

1. Partially solving $\pi_R^i = \arg\max_{\pi_R} J_R(\pi_R)$ and obtaining $\eta_i = J_R(\pi_R^i)$.
2. Solving the $i$th optimization problem:

$$\max_{\pi} J_{TG}(\pi, \alpha) \quad \text{subject to} \quad J_R(\pi) \geq \eta_i \tag{5}$$

To solve the constrained problem 5, we chose to use the dual lagrangian method, which has been demonstrated to work well in reinforcement learning problems (Tessler et al., 2018; Bhatnagar & Lakshmanan, 2012). Using the Lagrange duality, we transform the constrained problem into an unconstrained min-max optimization problem. Considering Eq. 5 as the primal problem, the corresponding dual problem is:

$$\min_{\lambda \geq 0} \max_{\pi} \left[ J_{TG}(\pi, \alpha) + \lambda \left( J_R(\pi) - \eta_i \right) \right] =$$

$$\min_{\lambda \geq 0} \max_{\pi} \left[ (1 + \lambda) J_{TG}(\pi, \frac{\alpha}{1+\lambda}) - \lambda \eta_i \right] \tag{6}$$

Where $\lambda$ is the lagrange multiplier. Full derivation can be found in appendix A. The resulting unconstrained optimization problem is compromised of two optimization problems. The inner one is solving for $\pi$, and since $\eta_i$ is independent of $\pi$ this optimization problem is equal to solving the combined objective 3 with an effective balancing coefficient of $\frac{\alpha}{1+\lambda}$. We can see that the value of $\alpha$ changes its role. While in the primal problem, it balanced the two objectives, now it is only part of the balancing term. Moreover, since $\lambda \geq 0$ yields $\alpha \geq \frac{\alpha}{1+\lambda} \geq 0$, then $\alpha$ can be seen as the upper bound on the effective balancing.

The second stage is to solve for $\lambda$. The dual function is always a convex function since it is the point-wise maximum of linear functions (Boyd et al., 2004). Therefore it can be solved with gradient descent without worrying about local minimums. The gradient of Eq. 6 with respect to the lagrangian multiplier $\lambda$ yields the following update rule:

$$\lambda_{new} = \lambda_{old} - \mu[J_R(\pi) - \eta_i] \tag{7}$$

Where $\mu$ is the step size for updating the Lagrange multiplier. See appendix A for full derivation. The resulting update rule is quite intuitive. If the policy that uses the teacher's reward term achieves higher environmental reward than the one trained without the teacher, then decrease $\lambda$. This, in return, will increase the effective coefficient, thus leading to more reliance on the teacher in the next iteration. If the policy trained without the teacher's guidance achieves a higher reward, then increase $\lambda$, decreasing the weight of the teacher's reward.

When utilizing Lagrange duality to solve a constrained optimization problem, it is necessary to consider the duality gap. The presence of a non-zero duality gap implies that the solution obtained from the dual problem serves only as a lower bound to the primal problem and does not necessarily provide the exact solution. Our analysis demonstrates that in the specific case under consideration, the duality gap is absent. For proofs of our propositions see Appendix A.

---

**Algorithm 1** Teacher Guided Reinforcement Learning (TGRL)

---

**Input**: $\lambda_{init}$, $\alpha$, $N_{\text{collect}}$, $N_{\text{update}}$, $\mu$ Initialize policies $\pi$ and $\pi_R$, $\lambda_0 \leftarrow \lambda_{init}$ $i = 1 \cdots$ Collect $N_{\text{collect}}$ new trajectories and add them to the replay buffer. $j = 1 \cdots N_{\text{update}}$ Sample a batch of trajectories from the replay buffer. Update $Q_R$ and $Q_E$. Update $\pi_R$ by maximizing $Q_R$ Update $\pi$ by maximizing $Q_R + \frac{\alpha}{1+\lambda}Q_E$ Estimate $J_R(\pi) - J_R(\pi_R)$ using Eq. 8 $\lambda_i \leftarrow \lambda_{i-1} + \mu[J_R(\pi) - J_R(\pi_R)]$ $\pi$

---

**Proposition 3.1.** *Suppose that the rewards function $r(s,a)$ and the cross-entropy term $H^X(\pi|\bar{\pi})$ are bounded. Then for every $\eta_i \in \mathbb{R}$ the primal and dual problems described in Eq. 5 and Eq. 6 has no duality gap. Moreover, if the series $\{\eta_i\}_{i=1}^\infty$ converges, then there is no dual gap also at the limit.*

As a result of proposition 3.1, by solving the dual problem, we get a solution for the primal problem. Notice that in the general case, the cross-entropy term can reach infinity when the support of the policies does not completely overlap, thus making our algorithm not comply with the assumptions stated above. As a remedy, we clip the value of the cross-entropy term and work with the bounded version.

## 3.2 IMPLEMENTATION

**Off-policy approach**: We implemented our algorithm using an off-policy actor-critic. This allows us to separate between data collection and policy learning, removing the need to collect data using both $\pi$ and $\pi_R$ at every iteration. As our objective is a compound of two terms, so is the Q value we try to maximize: $Q_{TG} = Q_R + \frac{\alpha}{1+\lambda}Q_E$. Where the first one, $Q_R$, represents the values of actions with respect to the environmental reward objective (Eq. 1). and the second one, $Q_E$, represents the values of actions with respect to the teacher's loss (Eq. 2). We found that using two separate critics for representing these Q functions leads to more stable training since the network's output does not need to change when $\lambda$ changes. We also use two actors, which correspond to our two policies $\pi$ and $\pi_R$ and optimize them by maximizing their relevant Q values. See Algorithm 1 for an outline of the algorithm and appendix B for details.

**Estimating the performance difference**: As part of the algorithm we update $\lambda$ using gradient descent. As shown before, the gradient of the dual problem with respect to $\lambda$ is the performance difference between the two policies, $J_R(\pi) - J_R(\pi_R)$. During the training, we need to estimate this quantity in order to take the gradient step. One option is to collect data using both policies and then use the trajectories as Monte-Carlo estimation of the cumulative reward. This method gives a good approximation when using a large number of trajectories. However, it requires a lot of interactions with the environments, reducing the sample efficiency of our algorithm in the process. Another option is to rely on the data we already have in our replay buffer for estimating this quantity. This data, however, were not collected using the current policies, and therefore we need to use value function approximations to make use of it. To achieve that we extended prior results from (Kakade & Langford, 2002; Schulman et al., 2015) known as the *objective difference lemma* to the off-policy case:

**Proposition 3.2.** *Let $\rho(s,a,t)$ be the distribution of states, actions, and timesteps currently in the replay buffer. Then the following is an unbiased approximation of the performance difference:*

$$J_R(\pi) - J_R(\pi_R) =$$

$$\mathbb{E}_{(s,a,t)\sim\rho}[\gamma^t(A_{\pi_R}(s,a) - A_\pi(s,a))] \tag{8}$$

Regardless of the method chosen for the estimation, one of the challenges of estimating the performance difference between the student and the teacher policies is the variability in the scale of the policies' performance across different environments and at different points in time. This makes it difficult to determine an appropriate learning rate for the weighting factor $\lambda$, which will work effectively in all settings. To address this issue, we found it necessary to normalize the performance difference value during the training process. This normalization allows us to use a fixed learning rate across all of our experiments.

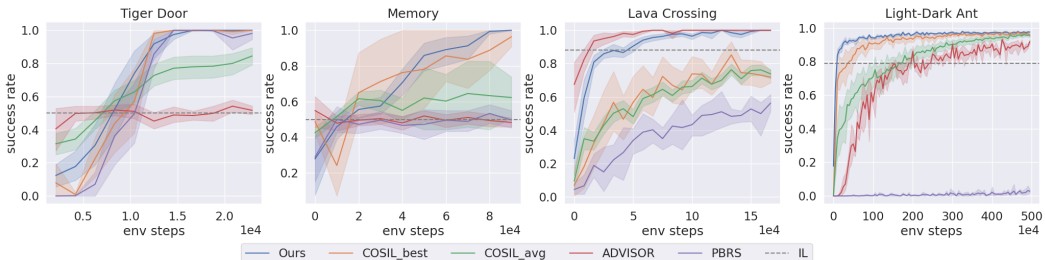

Figure 3: Success rate of various algorithm on domains with Imitation Gap. Overall, Our algorithm (blue) performs as well or better than competing methods across all tasks.

## 4 EXPERIMENTS

We perform three sets of experiments. In Sec. 4.1, we provide a comparison to previous work. In Sec. 4.2 we solve an object reorientation problem with tactile sensors, a difficult partial observable task that both RL and IL struggle to solve. In Sec. 4.2 we do ablations of our own method to show the effect of individual components.

### 4.1 TGRL PERFORMS WELL, WITHOUT A NEED FOR HYPERPARAMETER TUNING

Our goal in conducting the following experiments is twofold: (1) to showcase the robustness of TGRL with regard to the choice of its hyperparameters and (2) to check its ability to achieve competitive results when compared to other algorithms. To achieve that we will compare TGRL to the following algorithm:

**IL**. A pure imitation learning approach that tries to optimize only Eq. 2.

**COSIL**, from (Nguyen et al., 2022). This algorithm also uses entropy-augmented RL (Eq. 3) to combine the environmental reward and the teacher's guidance. To adjust the balancing coefficient $\alpha$, they propose an update rule that aims for a target cross-entropy between the student and the teacher. More formally, giving a target variable $\bar{D}$, they try to minimize $\alpha(J_E(\pi) - \bar{D})$ using gradient descent. Choosing a correct $\bar{D}$ is not a trivial task since we don't know beforehand how similar the student and the teacher should be. Moreover, even the magnitude of $\bar{D}$ can change drastically between environments, depending on the action space support. To tackle this issue, we run a hyperparameter search with $N = 8$ different values of $\bar{D}$ and report the performance of both the best and average values.

**ADVISOR-off**, an off-policy version of the algorithm from (Weihs et al., 2021). Instead of having a single coefficient to balance between the reward terms, this paper chose to create a state-dependent balancing coefficient. To do so, they first trained an auxiliary policy $\pi_{aux}$ using only imitation learning loss. Then, for every state, they compare the actions distribution of the teacher versus these of the auxiliary policy. The idea is that when the two disagree about the required action, it means that there is an information gap, and for this state, we should not trust the teacher. This is reflected in a coefficient that gives weight to the environmental reward.

**PBRS**, Concurrently and independently with our work (authors, 2022) propose to use potential-based reward shaping (PBRS) to mitigate the issue of imitation gap. PBRS, originated from (Ng et al., 1999), uses a given value function $V(s)$ to assign higher rewards to more beneficial states, which can lead the agent to trajectories favored by the policy associated with that value function:

$$r_{new} = r_{env} + \gamma V(s_{t+1}) - V(s_t)$$

where $r_{env}$ is the original reward from the environment. Since their algorithm is on-policy and we wanted to create a fair comparison, we created our own baseline based on the same approach. First, we train a policy $\pi^{IL}$ by minimizing only the imitation learning loss (Eq. 2). Then, we train a neural network to represent the value function of this policy. Using this value function, we augment the rewards using PBRS and train an agent using SAC over the augmented reward function.

In order to make a proper comparison, we perform experiments on a diverse set of domains with both discrete and continuous action spaces, proprioceptive and pixel observations:

**Tiger Door**. As described before, the task is to reach the goal cell without stepping on the failure cell. Pixel observations with discrete action space.

**Lava Crossing**. A minigrid environment where the agent starts in the top-left corner and needs to navigate through a maze of lava in order to get to the bottom-right corner. The episode ends in failure if the agent steps on the lava. The teacher has access to the whole map, while the student only sees a patch of 5x5 cells in front of it. Pixel observations with discrete action space.

**Memory**. The agent starts in a corridor containing two objects. It then has to go to a nearby room containing a third object, similar to one of the previous two. The agent's goal is to go back and touch the object it saw in the room. The episode ends in success if the agent goes to the correct object and in failure otherwise. While the student has to go to the room to see which object is the current one, the teacher starts with that knowledge and can go to it directly. Pixel observations with discrete action space.

**Light-Dark Ant**. A Mujoco Ant environment with a fixed goal and a random starting position. The starting position and the goal are located at the "dark" side of the room, where the agent has access only to a noisy measurement of its current location. It has to take a detour through the "lighted" side of the room, where the noise is reduced significantly, enabling it to understand its location. On the other hand, the teacher has access to its precise location at all times, enabling it to go directly to the goal. This environment is inspired by a popular POMDP benchmark (Platt Jr et al., 2010). Proprioceptive observation with continuous action space.

For a fair comparison, we used the same code and hyperparameters between across the various algorithms, changing only the algorithm-specific ones. While tuning the necessary hyperparameter for each algorithm, for TGRL we used only one value for the initial coefficient and coefficient learning rate for all environments, See appendix B for further details.

**Results**. The comparison results, depicted in Figure 3, demonstrate that TGRL exhibits comparable or superior performance across all tasks. Moreover, it achieved an almost perfect success rate across all environments, demonstrating that it is not suffering from the Imitation Gap as the IL method. While COSIL also demonstrates comparable performance when its hyperparameters are carefully tuned, the average performance across all hyperparameter configurations is significantly lower. This highlights its sensitivity to the choice of hyperparameters. The *PBRS* method also does not require hyperparameter tuning but has slower convergence than the other teacher-student methods. This comparison aligned with what has been demonstrated before by Cheng et al. (2021).

As can be seen, *ADVISOR* was able to solve some of the tasks successfully but completely failed on *Tiger Door* and *Lava Crossing*, instead converging to a sub-optimal policy similar to that of Imitation Learning method. This happens due to a fundamental limitation of the *ADVISOR* algorithm. As a reminder, the ADVISOR algorithm utilizes a state-based coefficient that compares an imitation learning policy to the teacher policy in order to determine the relative weighting between the IL loss and the environment reward loss. Looking at the *Tiger Door* environment, the first point where the policies differ is in the corridor split, but this is too late, as the divergence from the teacher should have occurred near the pink button. This problem will happen in every environment where the actions that should diverge from the teacher policy need to occur prior to the point where observation differences would force a different action.

To demonstrate the robustness of our algorithm to the choice of the initial coefficient value, we also performed a set of experiments in the *Lava Crossing* environment with different initial values. The results, depicted in Figure 4, indicate that the proposed algorithm can effectively adjust the coefficient value, regardless of the initial value, and converge to the optimal policy.

In order to ensure the versatility of our proposed algorithm, we also conducted experiments in environments without Imitation Gap. As the presence of an Imitation Gap is dependent on the specific task and the observations available to the agent, it is difficult to predict beforehand if such a gap exists in a given environment. The results of these experiments, presented in appendix C, demonstrate that our algorithm, TGRL, is able to effectively handle problems in environments without Imitation Gap, further solidifying its potential utility across a wide range of tasks.

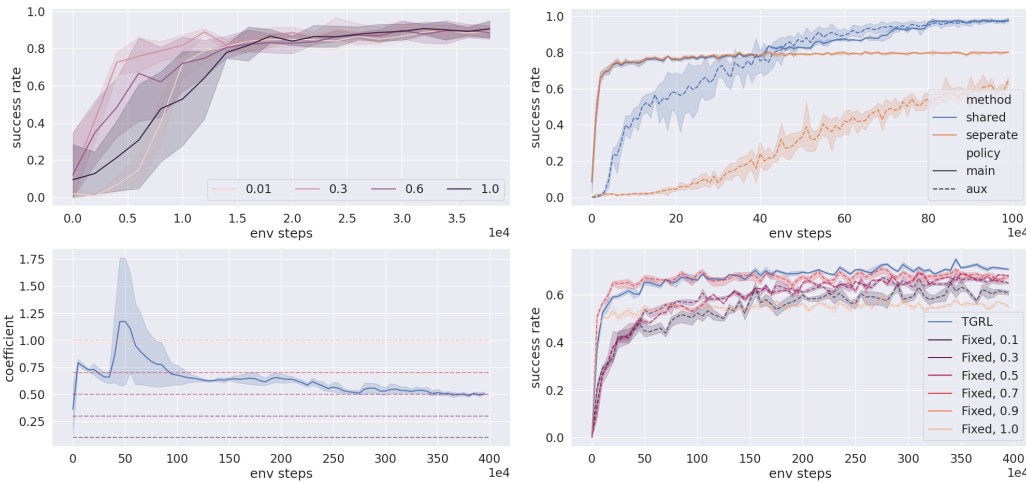

Figure 4: (Top-Left) Our algorithm performance in *Lava Crossing* is robust to different initial co-efficient values. (Top-Right) Convergence plots of the main policy $\pi$ and auxiliary policy $\pi_R$ in *Light-Dark Ant* with separate or joint replay buffers between them. (Bottom) Adaptive balancing coefficient between the teacher guidance and the RL loss provides better asymptotic convergence than a fixed coefficient in the *Shadow Hand* environment. On the right graph, there is the performance for different values and for TGRL. On the left graph, one can see the dynamics of the coefficient during the training.

## 4.2 TGRL CAN SOLVE DIFFICULT ENVIRONMENTS WITH SIGNIFICANT PARTIAL OBSERVABILITY.

We examined our algorithm's performance in handling heavily occluded observations of the environment's state, which are typically a challenge for reinforcement learning methods. To do so, we used the Shadow hand test suite with touch sensors (Melnik et al., 2021). In this task, the agent controls the joints of the hand's fingers and manipulates a pen to a target pose. The agent has access to its own current joint positions and velocity and controls them by providing desired joints angle to the hand's controller. Since we have access to the simulation, we were able to train a teacher policy that has access to the precise pose and velocity of the pen as part of its state. The student, however, has only access to an array of tactile sensors located on the palm of the hand and the phalanges of the fingers. The student needs to use the reading of these sensors to infer the pen's current pose and act accordingly.

While training our agents, we used a dense reward function that takes as a cost the distance between the current pen's pose and the goal. The pen has rotational symmetry, so the distance was taken only over rotations around the x and y axes. A trajectory was considered successful if the pen reached a pose of fewer than 0.1 radians than the goal pose.

The results of our experiments can be shown in figure 2. The performance shown is measured over a set of 1,000 randomly sampled poses. At first, we trained a teacher on the full state space using Soft Actor-Critic and Hindsight Experience Replay (HER) (Andrychowicz et al., 2017). The teacher achieved a 78% success rate after $5 \cdot 10^6$ environment steps. In parallel, using the same algorithms, we trained a history-dependent agent on the observation space (which includes only joint positions and the touch sensor's reading). This agent, which we will denote as the baseline agent, was only able to achieve a 47% success rate, and this is after $16 \cdot 10^6$ environments steps. This gap between the teacher and the baseline agent demonstrates the difficulty of solving this reorientation task based on tactile sensing only.

For the student, we compared using only imitation learning loss versus our algorithm. As can be seen from the graph, The imitation learning policy converged fast, but only to around 54% success rate. This is considerably less than the teacher's, due to the Imitation gap discussed before. The policy trained with our algorithm reached a 73% success rate, which is significantly higher. This demonstrates the usefulness of our algorithm and its ability to use the teacher's guidance while learning from the reward at the same time.

### 4.3 ABLATIONS

**Joint versus separate replay buffer**.Our algorithm uses two policies, the main policy $\pi$ and an auxiliary policy $\pi_R$. The goal of the auxiliary policy $\pi_R$ is to be used in the update rule of the Lagrange multiplier $\lambda$. During our experiments, we found that having a joint replay buffer between the policies is a must in order to achieve good performance. An example of that can be found in Figure 4, where we compare results in the *Light-Dark Ant* environment. We hypothesize that the reason is that our algorithm relies on the auxiliary policy $\pi_R$ to have a competitive performance to the main policy $\pi$ in order to adjust the coefficient between the teacher's guidance and the environmental reward. Having a single replay buffer for both policies helps in achieving a similar convergence rate since it enables each policy to learn from the successes and failures of the other.

**Fixed versus adaptive balancing coefficient**. One of the benefits of our method is the fact that the balancing coefficient between the combined objective (Eq. 3) is dynamic, changing during the training process based on the value of $\lambda$. To show that there are benefits of having an adaptive coefficient, we trained our algorithm with a set of fixed coefficients. The results, shown in Figure 4, highlight the advantage of our method. Not only that there is no need to search for the best parameter, but even if it can be found, it will lead to inferior results compared to a dynamic coefficient.

## 5 CONCLUSION AND FUTURE WORK

In this work, we examined the paradigm of using teacher-student methods to solve complicated POMDP. We presented an algorithm that dynamically balances between following the teacher and solving the environment using reinforcement learning loss, thus overcoming the limitation of prior methods. Although success remains to be demonstrated on real-world problems, we believe that our algorithm can aid in solving many tasks, especially those that exhibit significant partial observability, as we demonstrated with the robotic hand with tactile sensing domain.

An important investigation that we leave to future work is to have the balancing coefficient be state-dependent. As the difference between the teacher's and student's actions can depend on the specific state, we think this can lead to accelerated convergence compared to using one value for all states. However, how to dynamically update this value during the training process remains an open question.

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

## A   Derivations and Proofs

### A.1   Derivation of the Dual Problem

Given the Primal Problem we derived in Eq. 5:

$$\max_{\pi} J_{TG}(\pi, \alpha) \quad \text{subject to} \quad J_R(\pi) \geq \eta_i$$

The corresponding Lagrangian is:

$$\mathcal{L}(\pi, \lambda) = J_{TG}(\pi, \alpha) + \lambda \left( J_R(\pi) - \eta_i \right) =$$

$$\mathbb{E}_{\pi} \left[ \sum_{t=0}^{\infty} \gamma^t (r_t - \alpha H_t^X(\pi|\bar{\pi})) \right] + \lambda \mathbb{E}_{\pi} \left[ \sum_{t=0}^{\infty} \gamma^t r_t \right] - \lambda \eta_i =$$

$$\mathbb{E}_{\pi} \left[ \sum_{t=0}^{\infty} \gamma^t \left( (1+\lambda) r_t - \alpha H_t^X(\pi|\bar{\pi}) \right) \right] - \lambda \eta_i =$$

$$\mathbb{E}_{\pi} \left[ (1+\lambda) \sum_{t=0}^{\infty} \gamma^t \left( r_t - \frac{\alpha}{1+\lambda} H_t^X(\pi|\bar{\pi}) \right) \right] - \lambda \eta_i =$$

$$(1+\lambda) J_{TG}(\pi, \frac{\alpha}{1+\lambda}) - \lambda \eta_i$$

And therefore out Dual problem is:

$$\min_{\lambda \geq 0} \max_{\pi} \left[ (1+\lambda) J_{TG}(\pi, \frac{\alpha}{1+\lambda}) - \lambda \eta_i \right]$$

### A.2   Derivation of update rule for $\lambda$

The gradient of the dual problem with respect to $\lambda$ is:

$$\nabla_{\lambda} \left[ (1+\lambda) J_{TG}(\pi, \frac{\alpha}{1+\lambda}) - \lambda \eta_i \right] =$$

$$\nabla_{\lambda} \left[ \mathbb{E}_{\pi} \left[ \sum_{t=0}^{\infty} \gamma^t \left( (1+\lambda) r_t - \alpha H_t^X(\pi|\bar{\pi}) \right) \right] - \lambda \eta_i \right] =$$

$$\mathbb{E}_{\pi} \left[ \sum_{t=0}^{\infty} \gamma^t r_t \right] - \eta_i =$$

$$J_R(\pi) - \eta_i$$

### A.3   Duality Gap - Proof for Proposition 3.1

We start by restating our assumptions and discuss why they hold for our problem:

**Assumption A.1.** The rewards function $r(s, a)$ and the cross-entropy term $H^X(\pi|\bar{\pi})$ are bounded.

**Justification for A.1.** This is achieved by using a clipped version of the cross entropy term. We will add that we found the clipping helpful in practice since it stops this term from reaching infinity when the support of the teacher and the student action distributions are not the same.

**Assumption A.2.** The sequence $\{\eta_i\}_{i=1}^{\infty}$ is monotincally increasing and converging, i.e., there exist $\eta \in \mathbb{R}$ such that $\lim_{i \to \infty} \eta_i = \eta$.

**Justification for A.2.** We will remind that the sequence $\{\eta_i\}_{i=1}^{\infty}$ is the result of incrementally solving $\max_{\pi_R} J_R(\pi_R)$. Having this sequence be monotonically increasing is equivalent to a guarantee for policy improvement in each optimization step, an attribute of several RL algorithms such as Q-learning or policy gradient (Sutton & Barto, 2018). Regarding convergence, since the reward is upper bound from assumption A.1, then we have an upper bounded monotonically increasing sequence of real numbers, which is proved to converge.

**Assumption A.3.** There exist $\epsilon > 0$ such that for all $i$, the value of $\eta_i$ is at most $J_R(\pi^*) - \epsilon$.

**Justification for A.3.** This assumption is equivelant to stating that $J_R(\pi^*) - J_R(\pi_R) > 0$, meaning that $\pi_R$ is never optimal. Without further assumption on the algorithm used to optimize $\pi_R$, we can not guarantee that this will not happen. However, if it happens, it means that we were able to find the optimal policy, and therefore there is no need to continue with the optimization procedure. As a remedy, we will define a new sequence $\{\tilde{\eta}_i\}_{i=1}^{\infty}$ where $\tilde{\eta}_i = \eta_i - \epsilon$ and will use it instead of the original $\eta_i$. Since $\epsilon$ can be as small as we want, its effect on the algorithm is negligible and it served mainly for the completeness of our theory.

Before going into our proof, we will cite Theorem 1 of (Paternain et al., 2019), which is the basis of our results:

**Theorem A.4.** *Given the following optimization problem:*

$$P^* = \max_{\pi} \mathbb{E}_{\pi} \left[ \sum_{i=0}^{H} \gamma^t r_0(s_t, a_t) \right] \quad \textit{subject to}$$

$$\mathbb{E}_{\pi} \left[ \sum_{i=0}^{H} \gamma^t r_i(s_t, a_t) \right] \geq c_i, \quad i = 1...m,$$

*And its Dual form:*

$$D^* = \min_{\lambda \geq 0} \max_{\pi} \mathbb{E}_{\pi} \left[ \sum_{i=0}^{H} \gamma^t r_0(s_t, a_t) \right] +$$

$$\lambda \sum_{i=1}^{m} \left[ \mathbb{E}_{\pi} \left[ \sum_{i=0}^{H} \gamma^t r_i(s_t, a_t) \right] - c_i \right]$$

*suppose that $r_i$ is bounded for all $i = 0, ..., m$ and that Slater's condition holds. Then, strong duality holds, i.e., $P^* = D^*$.*

Having stated that, we will move to prove the two parts of our proposition:

**Proposition A.5.** *Given assumption A.1 and A.3, for every $\eta_i \in \mathbb{R}$, the constrained optimization problem Eq. 5 and its dual problem defined in Eq. 6 do not have a duality gap.*

*Proof.* We align our problem with Theorem A.4 notations by denote as follows:

$$r_0 : r_t - \alpha H_t^X, \quad r_1 : r_t, \quad c_1 : \eta_i$$

And we can see that our problem is a specific case of the optimization problem defined above. For every $\eta_i$, there is a set feasible solutions in the form of an $\epsilon$-neighborhood of $\pi^*$. This holds since $J_R(\pi^*) > J_R(\pi) - \epsilon$ for every $\pi \notin \pi^*$. Therefore, Slater's condition holds as it required that the feasible solution set will have an interior point. Together with assumption A.1, we have all that we need to claim that Theorem A.4 applies to our problem. Therefore, there is no duality gap. $\square$

**Proposition A.6.** *Given all our assumptions, the constrained optimization problem at the limit:*

$$\max_{\pi} J_{TG}(\pi, \alpha) \quad \textit{subject to} \quad J_R(\pi) \geq \eta$$

*has no duality gap.*

*Proof.* Our proof will be based on the Fenchel-Moreau theorem (Rockafellar, 1970) that states:

*If (i) Slater's condition holds for the primal problem and (ii) its perturbation function $P(\xi)$ is concave, then strong duality holds for the primal and dual problems.*

Denote $\eta_{\lim}$ the limit of the sequence. Without loss of generality, we assume that $\eta_{\lim} = J_R(\pi^*) - \epsilon$. If not, we will just adjust $\epsilon$ accordingly. As in the last proof, Slater's condition holds since there is a set of feasible policies for the problem. Regarding the second requirement, the sequence of perturbation functions for our problem is:

$$P(\xi) = \lim_{i \to \infty} P_i(\xi)$$

$$\text{where} \quad P_i(\xi) = \max_{\pi} J_{TG}(\pi, \alpha)$$

$$\text{subject to} \quad J_R(\pi) \geq \eta_i + \xi$$

Notice that this is a scalar function since $P_i(\xi)$ is the maximum objective itself, not the policy that induces it. We will now prove that this sequence of functions converges point-wise:

- For all $\xi > \epsilon$ we claim that $P(\xi) = \lim_{i \to \infty} P_i(\xi) = -\infty$. As a reminder $\eta_i$ converged to $J_R(\pi^*) - \epsilon$. It means that there exists $N$ such that for all $n > N$, we have $|\eta_n - J_R(\pi^*) + \epsilon| < \frac{\xi}{2} - \epsilon$. Moreover, since $J_R(\pi^*) - \epsilon$ is also the upper bound on the series of $\eta_i$ we can remove the absolute value and get:

$$0 \leq J_R(\pi^*) - \epsilon - \eta_n < \frac{\xi}{2} - \epsilon$$

  This yields the following constraint:

$$J_R(\pi_\theta) \geq \eta_n + \xi > J_R(\pi^*) - \frac{\xi}{2} + \xi = J_R(\pi^*) + \frac{\xi}{2}$$

  But since $\xi > \epsilon > 0$ and $\pi^*$ is the optimal policy, no policies are feasible for this constraint, so from the definition of the perturbation function, we have $P_n(\xi) = -\infty$. This holds for all $n > N$ and, therefore also $\lim_{i \to \infty} P_i(\xi) = -\infty$.

- For all $\xi \leq \epsilon$ we will prove convergence to a fixed value. First, we claim that the perturbation function has a lower bound. This is true since the reward function and the cross-entropy are bounded, and the perturbation function value is a discounted sum of them.
  In addition, the sequence of $P_i(\xi)$ is monotonically decreasing. To see it, remember that the sequence $\{\eta_i\}_{i=1}^{\infty}$ is monotonically increasing. Since $J_R(\pi)$ is also upper bounded by $J_R(\pi^*)$, then the feasible set of the $(i + 1)$ problem is a subset of the feasible set of the $i$ problem, and all those which came before. Therefore if the solution to the $i$ problem is still feasible it will also be the solution to the $i + 1$ problem. If not, then it has a lower objective (since it was also feasible in the $i$ problem), resulting in a monotonically decreasing sequence. Finally, for every $\eta_i$ there is at least one feasible solution, $J_R(\pi^*)$, meaning the perturbation function has a real value. To conclude, $\{P_i(\xi)\}_{i=1}^{\infty}$ is a monotonically decreasing, lower-bounded sequence in $\mathbb{R}$ in therefore it converged.

After we established point-wise convergence to a function $P(\xi)$, all that remain is to proof that this function is concave. According to proposition A.5, each optimization problem doesn't have a duality gap, meaning its perturbation function is concave. Since every function in the sequence is concave, and there is pointwise convergence, $P(\xi)$ is also concave. To conclude, from the Fenchel-Moreau theorem, our optimization problem doesn't have a duality gap in the limit. □

### A.4 PERFORMANCE DIFFERENCE ESTIMATION - PROOF FOR PROPOSITION 3

*Proposition:* Let $\rho(s, a, t)$ be the distribution of states, actions, and timesteps currently in the replay buffer. Then the following is an unbiased approximation of the performance difference:

$$J_R(\pi) - J_R(\pi_R) =$$

$$\mathbb{E}_{(s,a,t) \sim \rho}[\gamma^t (A_{\pi_R}(s, a) - A_\pi(s, a))]$$

*Proof:* Let $\pi_{RB}$ be the behavioral policy induced by the data currently in the replay buffer, meaning:

$$\forall s \in S \quad \pi_{RB}(a|s) = \frac{\sum_{a' \in RB(s)} I_{a'=a}}{\sum_{a' \in RB(s)} 1}$$

Using lemma 6.1 from (Kakade & Langford, 2002), for every two policies $\pi$ and $\tilde{\pi}$ We can write:

$$\eta(\tilde{\pi}) - \eta(\pi) = \eta(\tilde{\pi}) - \eta(\pi_{RB}) + \eta(\pi_{RB}) - \eta(\pi) =$$

$$-[\eta(\pi_{RB}) - \eta(\tilde{\pi})] + \eta(\pi_{RB}) - \eta(\pi) =$$

$$-\sum_{s}\sum_{t=0}^{\infty}\gamma^t P(s_t = s|\pi_{RB})\sum_{a}\pi_{RB}(a|s)A_{\tilde{\pi}}(s,a)+$$

$$\sum_{s}\sum_{t=0}^{\infty}\gamma^t P(s_t = s|\pi_{RB})\sum_{a}\pi_{RB}(a|s)A_{\pi}(s,a)=$$

$$\sum_{s}\sum_{t=0}^{\infty}\gamma^t P(s_t = s|\pi_{RB})\sum_{a}\pi_{RB}(a|s)[A_{\pi}(s,a)-A_{\tilde{\pi}}(s,a)]$$

Assuming we can sample tuples of $(s, a, t)$ from our replay buffer and denote this distribution $\rho_{RB}(s, a, t)$ we can write the above equation as:

$$\eta(\tilde{\pi}) - \eta(\pi) = \sum_{s,a,t}\rho_{RB}(s,a,t)\gamma^t[A_{\pi}(s,a)-A_{\tilde{\pi}}(s,a)]$$

Which we can approximate by sampling such tuples from the replay buffer.

## B    EXPERIMENTAL DETAILS

In this section, we outline our training process and hyperparameters.

Our algorithm optimizes two policies, $\pi$, and $\pi_R$, using off-policy Q-learning. The algorithm itself is orthogonal to the exact details of how to perform this optimization. For the discrete Gridworld domains (*Tiger Door*, *Memory* and *Lava Crossing*), we used DQN (Mnih et al., 2015) with soft target network updates, as proposed by (Lillicrap et al., 2015), which has shown to improve the stability of learning. For the rest of the continuous domains, we used SAC (Haarnoja et al., 2018) with the architectures of the actor and critic chosen similarly and with a fixed entropy coefficient. For both DQN and SAC, we set the soft target update parameter to 0.005. As was mentioned in the paper, we represent the Q function using to separate networks, one for estimating $Q_R$ and another for estimating $Q_E$. When updating a Q function, it has to be done with respect to some policy. We found that doing so with respect to policy $\pi$ yields stable performance across all environments.

For *Tiger Door*, *Memory*, and *Lava Crossing*, the teacher is a shortest-path algorithm executed over the grid map. For *Light-Dark Ant*, the teacher is a policy trained using RL over the privileged observation space until achieving a success rate of 100%. In all of our experiments, we average performance over 5 random seeds and present the mean and 95% confidence interval.

For all proprioceptive domains, we used a similar architecture across all algorithms. The architecture includes two fully-connected (FC) layers for embedding the past observations and actions separately. These embeddings are then passed through a Long Short-Term Memory (LSTM) layer to aggregate the inputs across the temporal domain. Additionally, the current observation is embedded using an FC layer and concatenated with the output of the LSTM. The concatenated representation is then passed through another fully-connected network with two hidden layers, which outputs the action. The architecture for pixel-based observations are the same, with the observations encoded by a Convolutional Neural Network (CNN) instead of FC. The number of neurons in each layer is determined by the specific domain. The rest of the hyperparameters used for training the agents are summarized in 5.

Our implementation is based on the code released by (Ni et al., 2022).

**Fair Hyperparameter Tuning.** We attempt to ensure that comparisons to baselines are fair. In particular, as part of our claim that our algorithm is more robust to the choice of its hyperparameters, we took the following steps. First, we re-implemented all baselines, and while conducting experiments, maintained consistent joint hyperparameters across the various algorithms. Second, all the experiments of our own algorithm, TGRL, used the same hyperparameters. We used $\alpha = 3$, initial $\lambda$ equal to 9 (and so the effective coefficient $\frac{\alpha}{1+\lambda} = 0.3$) and coefficient learning rate of 3e−3. Finally, for every one of the baselines we performed for each environment a search over all the algorithm-specific hyperparameters with N=8 different values for each one and report the best results (besides for COSIL, where we also report the average performance across hyperparameters).

| | Tiger Door | Lava Crossing | Memory | Light-Dark Ant | Shadow Hand |
|---|---|---|---|---|---|
| Max ep. length | 100 | 225 | 121 | 100 | 100 |
| Collected ep. per iter. | | 5 | | 10 | 120 |
| RL updates per iter. | | 500 | | 1000 | 1000 |
| Optimizer | | | Adam | | |
| Learning rate | | | 3e-4 | | |
| Discount factor ($\gamma$) | | | 0.9 | | |
| Batch size | | 32 | | 128 | 128 |
| LSTM hidden size | | 128 | | 256 | 128 |
| Obs. embedding | | 16 | | 32 | 128 |
| Actions embedding | | 16 | | 32 | 16 |
| Hidden layers after LSTM | | [128,128] | | [512,256] | [512, 256, 128] |

Figure 5: Hyperparameters table.

## C    ADDITIONAL RESULTS

Here we record additional results that were summarized or deferred in Section 4. In particular:

**Environments without Imitation Gap**. Determining the presence of an Imitation Gap in a given environment is a complex task, as it is dependent on the specific task and the observations available to the agent, which can vary significantly across different environments. As such, it can be challenging to know beforehand if an Imitation Gap exists or not. In the following experiment, we demonstrate that even in scenarios where an Imitation Gap does not exist, the use of our proposed TGRL algorithm yields results that are comparable to those obtained using traditional Imitation Learning (IL) methods, which are typically considered the best approach in such scenarios. This highlights the robustness and versatility of our proposed approach.

The experiment includes three classic POMDP environments from (Ni et al., 2022). These environments are a version of the Mujoco *Hopper*, *Walker2D*, and *HalfCheetah* environments, where the agent only have access to the joint positions but not to their velocities. The teacher, however, has access to both positions and velocities. As can be seen in Figure 6, TGRL converges a bit slower than IL but still manage to converge to the teacher's performance.

**Full training curves for Shadow Hand experiments.** In Figure 7, we provide the full version of the training curves that appears in Figure 2.

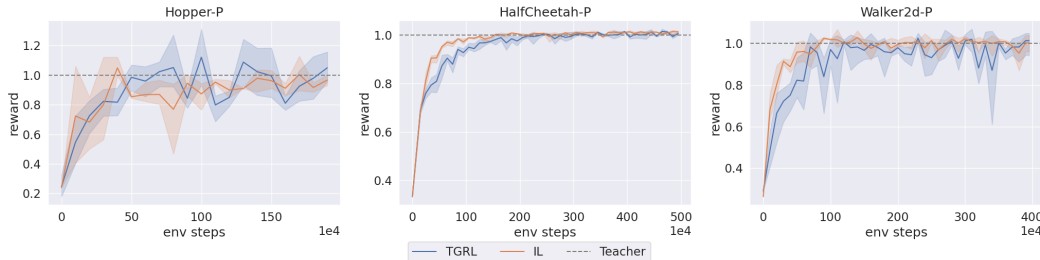

Figure 6: TGRL versus Imitation Learning on domains without Imitation Gap. The rewards are normalized based on the teachers' performance.

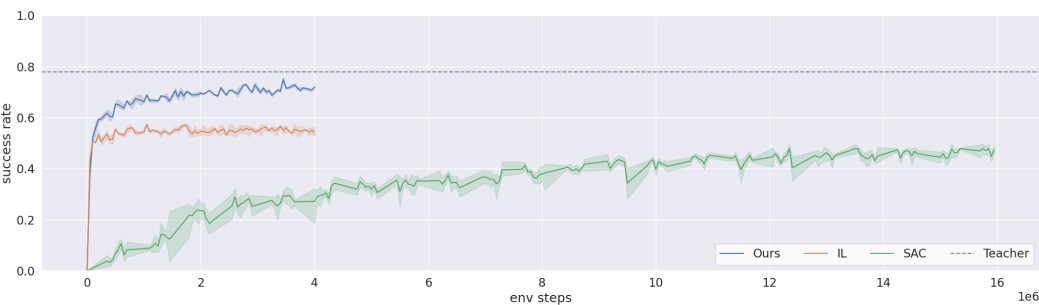

Figure 7: Full training curve of *Shadow Hand* pen reorientation with tactile sensors task

