# OpenReview forum: "TGRL: Teacher Guided Reinforcement Learning Algorithm for POMDPs"
_ICLR.cc/2023/Workshop/RRL — RRL 2023 Spotlight_

### Official Review · Reviewer_woZ4 · 2023-02-28

**Rating:** 4
**Confidence:** 4

**Review:**

This paper proposes a method to train student policies when a teacher policy with extra information is available. The proposed method combines two objectives, the usual reward and minimizing the cross-entropy between student and teacher polices, i.e. push the student to imitate the teacher. The novelty of the method is to adaptively set the weight applied to the imitation objective, this is done by viewing the problem as a constrained optimization where the policy must be at least as good as some other target policy. The target policy is set to be another parameterized policy, trained without extra information nor teacher guidance to maximize reward, and the weight is adapted via the Lagrangian dual.

This seems like a nice straightforward workshop contribution. The writing is good and I was able to understand the paper fairly well, and this seems like a good use of constrained optimization.

I do wonder if there is a simpler interpretation that exists that does not rely on access to a reference policy $\pi_R$; in some deeper sense, what this lagrangian adjustment captures is a desire to unconstrain the policy improvement step when the policy improvement direction has a "negative angle" to the policy imitation direction. I recall some papers (e.g. [1]) constraining gradient steps in a multitask setting by projecting the gradients onto the right subspace so as to avoid interference. Perhaps something similar could be interesting to the authors. I also agree with the authors that per-state weights would be a very interesting contribution.

[1] Gradient Surgery for Multi-Task Learning, Tianhe Yu, Saurabh Kumar, Abhishek Gupta, Sergey Levine, Karol Hausman, Chelsea Finn, 2020

---

### Official Review · Reviewer_CCzW · 2023-03-01
**Great work with strong experiments**

**Rating:** 4
**Confidence:** 4

**Review:**

The authors present a method that is able to learn from teacher demonstrations that may be able to act in a fully observable way and transfer this "teaching" to partially observable settings. They do so by introducing an additional constraint on the expected reward and optimize both for this reward and the policy in an iterative fashion. The paper has strong experiments, is well explained and concise, and should definitely be included!